# DEEP HYPERBOLIC HIERARCHICAL CLUSTERING

## ABSTRACT

Hierarchical clustering is a cornerstone of unsupervised learning, yet it has been a neglected method in modern deep learning. To enable deep hierarchical clustering, the unique geometry of hyperbolic space offers an ideal setting, renowned for its ability to embed tree-like structures with minimal distortion. However, prior attempts have been hampered by significant limitations, including geometric rigidity, a lack of scalability to large datasets, and imprecise formulations of key operations. This paper introduces a novel deep hyperbolic clustering framework that directly addresses these shortcomings through three key advancements. First, we present a generalized and rectified definition of the hyperbolic lowest common ancestor for both the Poincaré Ball and the Lorentz models of arbitrary curvature. Second, to address the critical issue of scalability, we employ a deep encoder that learns clusters in an exceptionally low-dimensional space compared to state of the art Euclidean methods. This makes our approach highly efficient and feasible for large-scale datasets. Finally, we introduce HoroPCA++, an improved and numerically stable dimensionality reduction technique for more faithful and lower distorted visualizations of the resulting hierarchies.

## 1 INTRODUCTION

Hierarchical clustering is a fundamental unsupervised task that organizes data into a nested hierarchy based on similarity. A key challenge in this domain is to produce hierarchies that are not only algorithmically sound but also meaningful, a quality often measured by objective functions like Dasgupta's cost by (Dasgupta, 2016). The function effectively penalizes a hierarchy if pairs of very similar points are only merged high up in the tree within a large sub-cluster. Therefore, minimizing this cost encourages the creation of hierarchies. The central problem we address is the optimization of this cost function within a continuous geometric framework, which offers a more flexible and powerful alternative to traditional discrete, combinatorial approaches.

The use of hyperbolic geometry for this task is particularly interesting and important because its geometric properties are intrinsically suited to represent hierarchical data. Unlike Euclidean space, which exhibits polynomial volume growth, hyperbolic space grows exponentially, allowing it to embed tree-like structures with significantly lower distortion (Sarkar, 2012; Bowditch, 2006; Gromov, 1987). This inherent advantage leads to more efficient and compact data representations, which can mitigate overfitting and provide a powerful inductive bias for a variety of learning tasks. By improving methods for clustering within this space, we can unlock more faithful and robust ways to discover latent hierarchical structures in complex datasets.

Working with a hyperbolic space is challenging. The geometric properties that make hyperbolic space so powerful also make it difficult to work with. Naive approaches designed for Euclidean space fail because they do not account for the principles of non-Euclidean geometry. Fundamental operations, such as finding the Lowest Common Ancestor (LCA) of two points, are not straightforward and require complex calculations involving geodesics and non-linear projections. Furthermore, implementing these operations is fraught with practical difficulties, including numerical instability that can arise from repeated computations and the risk of inverting near-singular matrices during optimization.

Although the seminal work of (Chami et al., 2020) introduced a continuous relaxation of Dasgupta's cost in hyperbolic space, this framework suffers from several critical limitations that have prevented its widespread application. First, previous solutions are geometrically rigid, being restricted to the Poincaré Ball model and assuming a fixed curvature. Second, they lack scalability, making them

impractical for large datasets common in modern machine learning. Third, their formulation of the hyperbolic LCA is geometrically imprecise, necessitating a formal correction. Finally, existing visualization techniques like HoroPCA (Chami et al., 2021) are not generalized to arbitrary curvatures and suffer from numerical instabilities. Our work differs by directly addressing these four shortcomings, proposing a framework that is more general, scalable, and theoretically sound.

Our approach is built on three key technical advancements:

- First, we rectify and generalize the definition of the hyperbolic LCA that is applicable to both the Poincaré Ball and Hyperboloid model across arbitrary curvatures, unifying these geometric settings.
- Second, to address scalability, we develop a deep clustering method that learns latent similarity representations using a contrastive encoder, enabling our framework to process large-scale datasets efficiently.
- Third, we present HoroPCA++, a numerically stable and generalized visualization technique to produce faithful low-distortion embeddings of the learned hierarchies.

Empirically, our method achieves state-of-the-art clustering performance on benchmark datasets, outperforming prior work on metrics such as Dasgupta's cost, Dendrogram Purity, Normalized Mutual Information (NMI) and Adjusted Rand Index (ARI).

## 2 RELATED WORK

**Hyperbolic Representation Learning.** Hyperbolic space provides a natural geometric framework for representing hierarchical data. Its defining characteristic of exponential volume growth enables low-distortion embeddings of tree-like structures even in two dimensions (Sala et al., 2018; Chepoi et al., 2012). This exponential scaling directly mirrors the branching structure of hierarchical data, where the number of nodes grows exponentially with tree depth (Sarkar, 2012; Bowditch, 2006; Gromov, 1987). In contrast, Euclidean space only exhibits polynomial volume growth, creating a fundamental mismatch with hierarchical structures that leads to a high distortion when embedding trees (Linial et al., 1995). Thus, hyperbolic space offers several computational advantages: embeddings require fewer dimensions to preserve structural relationships, reduced model complexity, mitigation of overfitting, and its intrinsic geometry promotes hierarchies without explicit architectural constraints. Building on this foundation, our work introduces a self-contained deep clustering algorithm that directly leverages these geometric benefits of hyperbolic space.

**Hierarchical Clustering in Hyperbolic Space.** The foundational work by (Chami et al., 2020) introduced a continuous relaxation of Dasgupta's cost, enabling continuous optimization of hierarchical clustering objectives within deep learning frameworks. However, this approach suffers from three fundamental limitations that restrict its practical applicability. First, the method exhibits geometric rigidity by constraining optimization to the Poincaré Ball with fixed curvature, limiting its flexibility across diverse data manifolds. Second, the proposed architecture suffers from severe scalability constraints, as the network complexity scales linearly with the size of the dataset $\mathcal{O}(nd)$, rendering it computationally infeasible for modern large-scale datasets prevalent in computer vision, natural language processing, and network analysis. Third, the original definition of the hyperbolic lowest common ancestor is geometrically imprecise, necessitating a formal rectification. Our work addresses these critical shortcomings through a novel scalable deep clustering framework that incorporates: (i) the rectified LCA formulation, (ii) support for multiple hyperbolic space models with arbitrary curvatures, and (iii) computational efficiency suitable for large-scale applications.

Another distinct, feature-based approach was proposed by (Monath et al., 2019), where each node in the hierarchy is represented by a continuous vector, and parent-child links are formed by minimizing a child-parent dissimilarity function. In contrast, our work follows the similarity-based paradigm of (Chami et al., 2020), optimizing a global cost function over leaf embeddings rather than learning explicit routing decisions.

**Deep Clustering in Euclidean Space.** Our approach to scalability is inspired by the paradigm of Deep Clustering (DC), which has become central to modern unsupervised learning. The core idea is to jointly optimize a deep neural network, such as an autoencoder, for representation learning alongside a clustering-specific loss function. This process encourages the network to learn a "cluster-

friendly" latent space in which data points belonging to the same group are drawn closer together. Foundational methods like DEC (Xie et al., 2016), IDEC (Guo et al., 2017), and DCN (Yang et al., 2017) demonstrated the power of this approach in Euclidean space. However, these methods are designed exclusively for Euclidean geometry and cannot be directly applied to hyperbolic space. Our work is the first to integrate a modern, autoencoder-based DC framework with the geometric principles of hyperbolic hierarchical clustering, thereby solving the critical scalability problem of prior hyperbolic methods.

**Hyperbolic Dimensionality Reduction.** Visualizing hyperbolic latent spaces requires specialized dimensionality reduction techniques that respect the geometry of the hyperbolic space. HoroPCA (Chami et al., 2021) is a widely adopted method that extends principal component analysis to hyperbolic geometries. Unlike other hyperbolic PCA techniques such as Principal Geodesic Analysis (Fletcher et al., 2004) or Barycentric Subspace Analysis (Pennec, 2018) that rely on geodesic projections, HoroPCA projects points along horospheres – surfaces in hyperbolic space that are locally isometric to Euclidean planes. Horospherical projections preserve distances between points, resulting in low-dimensional representations with reduced distortion. However, HoroPCA is limited to specific curvatures and exhibits numerical instabilities in practice. To address these limitations, we introduce HoroPCA++, a novel numerically robust extension that generalizes to arbitrary curvatures.

## 3 BACKGROUND

### 3.1 HYPERBOLIC SPACE

Our work employs two canonical models of hyperbolic space: the *Poincaré Ball* and the *Hyperboloid*. While these models capture the same underlying hyperbolic geometry, they provide distinct representations and offer unique views. The two isometrically equivalent models are mathematically defined as simply-connected Riemannian manifolds $(\mathcal{M}, g^{\mathcal{M}})$ with constant negative sectional curvature $-c$, where $c \in \mathbb{R}_{>0}$. This negative curvature fundamentally distinguishes hyperbolic geometry from both Euclidean (zero curvature) and spherical (positive curvature) geometries. For ease of exposition, we omit sub- and super-scripts denoting the dimensionality, curvature, etc., whenever they can be inferred from the context.

**Poincaré Ball.** The $n$-dimensional *Poincaré Ball* $(\mathbb{P}_c^n, g^{\mathbb{P}})$ is defined as the open ball of radius $1/\sqrt{c}$ centered at the origin $\mathbb{P}_c^n = \{ \boldsymbol{x} \in \mathbb{R}^n : ||\boldsymbol{x}||^2 < 1/c \}$, endowed with the Riemannian metric $g_{\boldsymbol{x}}^{\mathbb{P}} = \frac{2}{1-c\,||\boldsymbol{x}||^2} g_{\boldsymbol{x}}^{\mathbb{E}}$, where $g_{\boldsymbol{x}}^{\mathbb{E}}$ denotes the standard Euclidean metric tensor. This conformal relationship to Euclidean space can also be expressed by the family of inner products $\left\{ \langle \boldsymbol{u}, \boldsymbol{v} \rangle_{\boldsymbol{x}} : \mathcal{T}_{\boldsymbol{x}} \mathbb{P}_c^n \times \mathcal{T}_{\boldsymbol{x}} \mathbb{P}_c^n \to \mathbb{R}, \ (\boldsymbol{u}, \boldsymbol{v}) \mapsto \frac{2\langle \boldsymbol{u}, \boldsymbol{v} \rangle}{1-c\,||\boldsymbol{x}||^2} = \lambda_{\boldsymbol{x}}^c \langle \boldsymbol{u}, \boldsymbol{v} \rangle \right\}_{\boldsymbol{x} \in \mathbb{P}}$ induced by the metric. Here, $\mathcal{T}_{\boldsymbol{x}} \mathbb{P}_c^n$ denotes the tangent space with base point $\boldsymbol{x} \in \mathbb{P}_c^n$ and $\langle ., . \rangle$ the standard Euclidean dot product.

**Hyperboloid.** The $n$-dimensional *Hyperboloid* model $(\mathbb{H}_c^n, g_{\mathbb{H}})$ is defined as the forward sheet of a two-sheeted hyperboloid embedded within the $(n+1)$-dimensional Minkowski space $\mathbb{R}^{1,n}$ with $\mathbb{H}_c^n = \{ (x_0, \ldots, x_n) \in \mathbb{R}^{n+1} : \langle \boldsymbol{x}, \boldsymbol{x} \rangle_{\mathcal{L}} = -1/c, \ x_0 > 0 \}$. That is, the ambient space is the $(n+1)$-dimensional Minkowski space $\mathbb{R}^{1,n}$, which is a real vector space $\mathbb{R}^{n+1}$ endowed with the Minkowski inner product, an indefinite bilinear form defined as $\langle \boldsymbol{x}, \boldsymbol{y} \rangle_{\mathcal{L}} = -x_0^2 + x_1^2 + \cdots + x_n^2$ for $\boldsymbol{x}, \boldsymbol{y} \in \mathbb{R}^{n+1}$. The Riemannian metric on $\mathbb{H}_c^n$ is then obtained by restricting $\langle ., . \rangle_{\mathcal{L}}$ to the tangent spaces of the Hyperboloid, yielding a positive-definite metric as required for a Riemannian manifold.

**Hyperbolic Encoding and Decoding.** To retrieve hyperbolic latent representations from Euclidean ones, we employ a two-step encoding process that maps Euclidean vectors to the hyperbolic manifold $\mathcal{M}$. Given an input vector $\boldsymbol{v} \in \mathbb{R}^n$, we first apply a projection map $\phi$ to view $\boldsymbol{v}$ as a tangent vector of the manifold's origin $\bar{\boldsymbol{0}}$, and then use the exponential map $\exp_{\bar{\boldsymbol{0}}}$ to project it onto $\mathcal{M}$: $\mathbb{R}^n \xrightarrow{\phi} \mathcal{T}_{\bar{\boldsymbol{0}}} \mathcal{M} \xrightarrow{\exp_{\bar{\boldsymbol{0}}}} \mathcal{M}$. The *exponential map* $\exp_{\bar{\boldsymbol{0}}} : \mathcal{T}_{\bar{\boldsymbol{0}}} \mathcal{M} \to \mathcal{M}$ maps tangent vectors $\boldsymbol{v} \in \mathcal{T}_{\bar{\boldsymbol{0}}} \mathcal{M}$ to points on the manifold $\mathcal{M}$ such that the parametric curve $t \in [0, 1] \mapsto \exp_{\bar{\boldsymbol{0}}}(t\boldsymbol{v})$ traces the unique geodesic (shortest path) connecting the manifold's origin $\bar{\boldsymbol{0}}$ and $\exp_{\bar{\boldsymbol{0}}}(\boldsymbol{v})$.

In order to recover Euclidean latent representations from hyperbolic ones, we reverse the hyperbolic encoding operations through the inverse mappings. That is, we first apply the logarithmic map $\log_{\bar{\boldsymbol{0}}}$ to project points from the hyperbolic manifold back to the tangent space, followed by the

inverse projection $\phi^{-1}$ to recover Euclidean vectors: $\mathcal{M} \xrightarrow{\log_{\bar{\mathbf{0}}}} \mathcal{T}_{\bar{\mathbf{0}}}\mathcal{M} \xrightarrow{\phi^{-1}} \mathbb{R}^n$. The *logarithmic map* $\log_{\bar{\mathbf{0}}}: \mathcal{M} \to \mathcal{T}_{\bar{\mathbf{0}}}\mathcal{M}$ is the local inverse to $\exp_{\bar{\mathbf{0}}}$ and maps the points $\boldsymbol{x} \in \mathcal{M}$ from the manifold back to the tangent space $\mathcal{T}_{\bar{\mathbf{0}}}\mathcal{M}$, by finding $\boldsymbol{v} \in \mathcal{T}_{\bar{\mathbf{0}}}\mathcal{M}$ such that $\exp_{\bar{\mathbf{0}}}(\boldsymbol{v}) = \boldsymbol{x}$. The explicit formulas for these operations in both the Poincaré Ball and Hyperboloid models are provided in the Appendix A.7.

## 3.2 Hierarchical Clustering in Hyperbolic Space

**Dasgupta's Discrete Hierarchical Clustering Cost.** Hierarchical clustering organizes data into tree structures that capture meaningful relationships at multiple scales. To this aim, Dasgupta (Dasgupta, 2016) introduced a formal measure to quantify the quality of a hierarchy $T$ for a given set of points and their pairwise similarities $w$. It is defined as: $C_{\text{Dasgupta}}(T; w) = \sum_{i,j} w_{ij} |\text{leaves}(T[i \vee j])|$. The objective encourages similar points (high $w_{ij}$) to be clustered together early in the hierarchy, placing them closer to the leaves and farther from the root than dissimilar points, resulting in a small subtree size with a small number of leaves $|\text{leaves}(T[i \vee j])|$ for their Lowest Common Ancestor (LCA) $i \vee j$.

**Continuous Relaxation in Hyperbolic Space.** To make Dasgupta's cost optimizable, (Wang & Wang, 2018) first proposed an equivalent formulation based on triplets, which we present in Appendix A.6. Building on this, (Chami et al., 2020) developed a continuous, differentiable relaxation by leveraging the natural tree-like structure of hyperbolic space. The core idea is to replace discrete LCA operations with continuous hyperbolic analogues and non-differentiable functions such as the indicator function $\mathbb{1}$ with differentiable ones. Chami's hyperbolic hierarchical clustering objective is:

$$C_{\text{Chami}}(Z; w, \tau) = \sum_{i,j,k} \left( w_{ij} + w_{ik} + w_{jk} - \left\langle \begin{pmatrix} w_{ij} \\ w_{ik} \\ w_{jk} \end{pmatrix}, \sigma_\tau \begin{pmatrix} d_{\bar{\mathbf{0}}}(\boldsymbol{z_i} \vee \boldsymbol{z_j}) \\ d_{\bar{\mathbf{0}}}(\boldsymbol{z_i} \vee \boldsymbol{z_k}) \\ d_{\bar{\mathbf{0}}}(\boldsymbol{z_j} \vee \boldsymbol{z_k}) \end{pmatrix} \right\rangle \right) + 2 \sum_{i,j} w_{ij} \quad (1)$$

where $Z = \{\boldsymbol{z_1}, \ldots, \boldsymbol{z_n}\}$ are the hyperbolic embeddings of the data, $w_{ij}$ is the pairwise similarity of $\boldsymbol{z_i}$ and $\boldsymbol{z_j}$, $d_{\bar{\mathbf{0}}}(\boldsymbol{z_i} \vee \boldsymbol{z_j})$ is the distance of the hyperbolic LCA $\boldsymbol{z_i} \vee \boldsymbol{z_j}$ to the manifold's origin $\bar{\mathbf{0}}$ (Table 10), and $\sigma_\tau$ is the temperature-scaled softmax function with parameter $\tau$. This hyperbolic formulation can be optimized with standard gradient-based methods and yields $(1 + \varepsilon)$-optimal minimizers with respect to Dasgupta's discrete hierarchical clustering cost.

## 4 Deep Hyperbolic Clustering

### 4.1 Model Architecture

As illustrated in Figure 1, we use an autoencoder architecture for our experiments on image datasets. The encoder transform Euclidean inputs into hyperbolic latent representations following our hyperbolic encoding schema (3.1). The Euclidean decoder reverses this process to reconstruct the output and does not employ any specialized hyperbolic layers.

In our architecture, we employ established hyperbolic linear layers to process representations within the hyperbolic space. Specifically, we use a Poincaré Ball linear layer and a Hyperboloid model linear layer. These layers generalize standard affine transformations to their respective hyperbolic geometries, ensuring that outputs remain on the manifold. The full mathematical formulations for these layers are provided in the appendix in SectionA.4.1.

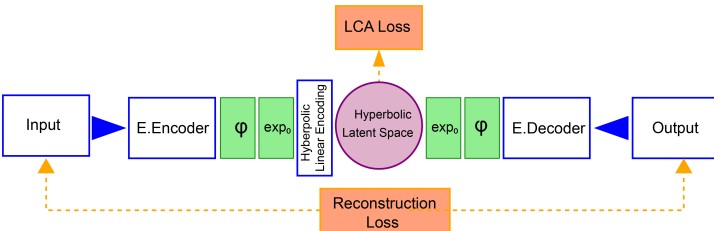

Figure 1: Model Architecture

## 4.2 Hyperbolic Lowest Common Ancestors

The concept of LCAs in trees has a natural analogue in hyperbolic geometry that exploits the tree-like structure of hyperbolic space. However, existing definitions require rectification to ensure consistency with the discrete case.

**Limitations of Existing Hyperbolic LCA Definitions.** Previous work by Chami (Chami et al., 2020) defined the hyperbolic LCA of two points $x, y \in \mathcal{M}$ as the point on the geodesic $\Gamma_{x,y}$ that is closest to the manifold's origin $\bar{0}$. This closest point is obtained through the geodesic orthogonal projection $\pi_\Gamma(\bar{0})$ of the origin $\bar{0}$ onto the geodesic. While geometrically intuitive, this definition exhibits a critical flaw. It fails to properly handle cases where one point is an ancestor of the other in the implied tree structure. In such cases, the orthogonal projection falls outside the geodesic segment connecting the two points, violating the fundamental property that LCAs should lie between their descendants as illustrated by Figure 2.

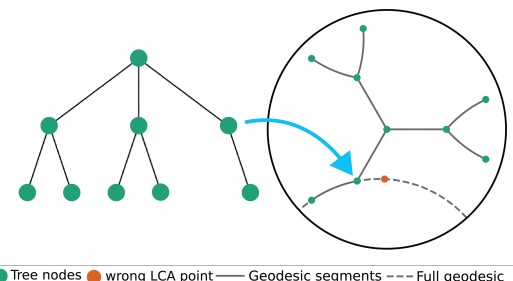

Figure 2: Illustration of the flaw in the previous hyperbolic LCA definition by (Chami et al., 2020). While the "wrong LCA point" (orange) lies on the full geodesic (dashed line), it falls outside the geodesic segment (solid line) connecting its two descendants, a problem corrected by the rectified LCA definition proposed in this work.

● Tree nodes ● wrong LCA point —— Geodesic segments --- Full geodesic

**Rectified Hyperbolic LCA Definitions.** To address this limitation, we propose a rectified definition that ensures the hyperbolic LCA lies on the geodesic segment $\gamma_{x,y} \subset \Gamma_{x,y}$ between points $x$ and $y$. This rectification preserves the essential properties of tree-based LCAs while maintaining geometric consistency.

**Definition 4.1.** Let $x, y$ be two points in either hyperbolic space model $\mathcal{M}$, and let $p = \pi_\Gamma(\bar{0})$ denote the geodesic orthogonal projection of the origin $\bar{0}$ onto the geodesic $\Gamma_{x,y}$. The hyperbolic lowest common ancestor $x \vee y$ is defined as:

$$
x \vee y = \begin{cases} \bar{0}, & \text{if } \cos(\angle_{x\bar{0}y}) = -1 \\ \arg\min_{z \in \{x,y\}} d_{\bar{0}}(z), & \text{if } \cos(\angle_{x\bar{0}y}) > \cos(\angle_{x\bar{0}p}) \text{ or } \cos(\angle_{x\bar{0}y}) > \cos(\angle_{y\bar{0}p}) \text{ or} \\ & \quad \cos(\angle_{x\bar{0}y}) = 1 \\ p, & \text{otherwise} \end{cases}
$$

The three cases in this definition correspond to distinct geometric configurations. The first case handles antipodal points, where the LCA is naturally the manifold's origin. The second case identifies ancestor-descendant relationships by comparing angular distances, ensuring the LCA is the closest ancestor rather than a point outside the connecting segment. The third case applies the standard orthogonal projection when both points lie on the same side of the origin and the projection falls within their connecting segment. In the following propositions, whose proofs are stated in Appendix A.2, we summarize the geodesic orthogonal projection $\pi_\Gamma(\bar{0})$ in the respective manifolds, which serves as input to the rectified LCA definition (4.1).

**Proposition 4.2.** *Let* $x, y \in \mathbb{P}^n_c$ *be two points in the Poincaré Ball that are not collinear. The geodesic orthogonal projection of the origin onto the geodesic* $\Gamma_{x,y}$ *connecting* $x$ *and* $y$ *is given by:*

$$
\pi_\Gamma(\bar{0}) = \left(1 - \frac{\sqrt{||z||^2 - \frac{1}{c}}}{||z||}\right) z, \quad where \tag{2}
$$

$$
z = \frac{\left[(1 + c||y||^2)\langle x, y\rangle - (1 + c||x||^2)||y||^2\right] x + \left[(1 + c||x||^2)\langle x, y\rangle - (1 + c||y||^2)||x||^2\right] y}{2c(|\langle x, y\rangle|^2 - ||x||^2||y||^2)}.
$$

**Proposition 4.3.** *Let $\boldsymbol{x}, \boldsymbol{y} \in \mathbb{H}_c^n$ be two points in the Hyperboloid. The geodesic orthogonal projection of the origin onto the geodesic $\Gamma_{\boldsymbol{x},\boldsymbol{y}}$ connecting $\boldsymbol{x}$ and $\boldsymbol{y}$ is given by:*

$$\boldsymbol{\pi}_\Gamma(\bar{\boldsymbol{0}}) = \frac{\alpha\boldsymbol{x} + \beta\boldsymbol{y}}{\sqrt{-c\langle\alpha\boldsymbol{x} + \beta\boldsymbol{y}, \alpha\boldsymbol{x} + \beta\boldsymbol{y}\rangle_{\mathcal{L}}}}, \quad where \tag{3}$$

$$\alpha = \frac{\sqrt{c}\,(x_0 + c\langle\boldsymbol{x}, \boldsymbol{y}\rangle_{\mathcal{L}}\,y_0)}{1 - (c\langle\boldsymbol{x}, \boldsymbol{y}\rangle_{\mathcal{L}})^2}, \beta = \frac{\sqrt{c}\,(y_0 + c\langle\boldsymbol{x}, \boldsymbol{y}\rangle_{\mathcal{L}}\,x_0)}{1 - (c\langle\boldsymbol{x}, \boldsymbol{y}\rangle_{\mathcal{L}})^2}.$$

### 4.3 Loss Function

Our models are trained end-to-end by jointly optimizing two objectives (Figure 1). Our total loss function combines a reconstruction loss with a hierarchical clustering loss: $\mathcal{L}_{\text{total}} = \mathcal{L}_{\text{recon}} + \lambda\mathcal{L}_{\text{LCA}}$, where $\mathcal{L}_{\text{recon}}$ is the autoencoder's reconstruction loss, $\mathcal{L}_{\text{LCA}}$ is a normalized version of the hyperbolic hierarchical clustering loss (1), and $\lambda \in \mathbb{R}_{>0}$ is a balancing hyperparameter.

**Reconstruction Loss.** $\mathcal{L}_{\text{recon}}$ ensures that the model learns high-quality latent representation by minimizing discrepancies between the inputs and their reconstructions, measured by the *Mean Squared Error*.

**Hierarchical Clustering Loss.** $\mathcal{L}_{\text{LCA}}$ constitutes the core of our approach, organizing embeddings within the hyperbolic latent space according to their hierarchical relationships. This loss builds upon the continuous, differentiable relaxation of Dasgupta's cost function 1. We define pairwise similarity between datapoints as: $w_{ij} = \frac{1}{1+d_{\mathcal{M}}(\boldsymbol{z}_i, \boldsymbol{z}_j)}$, where $d_{\mathcal{M}}(\boldsymbol{z}_i, \boldsymbol{z}_j)$ denotes the pairwise geodesic distance between the embedded tree nodes $i$ and $j$.

**Triplet Sampling.** Direct implementations of $\mathcal{L}_{\text{LCA}}$ require evaluating all possible triplets $(i, j, k)$ in the dataset, resulting in $\mathcal{O}(n^3)$ computational complexity for $n$ datapoints per optimization step. Even for moderate batch sizes this disagrees with our scalability objectives. Thus, we adopt a triplet sampling strategy, following prior work (Chami et al., 2020), that provides an unbiased stochastic estimate of the global objective $\mathcal{L}_{\text{LCA}}$ while maintaining computational efficiency.

**Loss Normalization.** To facilitate stable training dynamics and enable fair comparisons across datasets with varying scales, we normalize the hierarchical clustering loss $\mathcal{L}_{\text{LCA}}$ to $[0, 1]$:

$$\mathcal{L}_{\text{norm}} = \frac{\mathcal{L}_{\text{total}} - \mathcal{L}_{\text{min}}}{\mathcal{L}_{\text{max}} - \mathcal{L}_{\text{min}}}, \quad \text{where} \tag{4}$$

$$\mathcal{L}_{\text{min}} = \sum_{i,j,k} \min(w_{ij} + w_{ik}, w_{ij} + w_{jk}, w_{ik} + w_{jk}) + 2\sum_{i,j} w_{ij}$$
$$\mathcal{L}_{\text{max}} = \sum_{i,j,k} \max(w_{ij} + w_{ik}, w_{ij} + w_{jk}, w_{ik} + w_{jk}) + 2\sum_{i,j} w_{ij}$$

## 5 HoroPCA++

This section introduces **HoroPCA++**, a numerically robust dimensionality reduction method that extends the original HoroPCA (Chami et al., 2021) to hyperbolic space models with arbitrary curvatures. Our method addresses critical computational challenges while significantly broadening applicability across different hyperbolic geometries. For a comprehensive background on horospherical projections and theoretical foundations, we refer readers to the original HoroPCA paper. Here, we focus on the novel algorithmic contributions and improvements of HoroPCA++.

**Generalized Ideal Points.** Like its predecessor, HoroPCA++, starts by selecting $k$ principal components represented by ideal points $\boldsymbol{q_1}, \ldots, \boldsymbol{q_k}$. These points at infinity with respect to the hyperbolic space model correspond to directional vectors that parametrize the low dimensional target submanifold. To accommodate arbitrarily curved hyperbolic spaces, we formalize ideal points as follows.

**Definition 5.1.** For the Poincaré Ball, the set of ideal points $\boldsymbol{q} \in \mathcal{I}_{\mathbb{P}}^{n-1}$ is defined as its topological boundary $\mathcal{I}_{\mathbb{P}}^{n-1} = \left\{\boldsymbol{q} \in \mathbb{R}^n : \|\boldsymbol{q}\|^2 = 1/c\right\}$. For the Hyperboloid, the set of ideal points $\mathcal{I}_{\mathbb{H}}^{n-1}$ is defined as its asymptotic null cone $\mathcal{I}_{\mathbb{H}}^{n-1} = \left\{\boldsymbol{q} \in \mathbb{R}^{n+1} : \langle\boldsymbol{q}, \boldsymbol{q}\rangle_{\mathcal{L}} = 0\right\}$.

In the Hyperboloid model, ideal points represent specific directions rather than individual vectors, making them independent of the curvature $c$. However, while the ideal points themselves are curvature-independent in this representation, the horospherical projection operations remain curvature-dependent, necessitating our generalized approach. During optimization, these ideal points are iteratively refined to maximize the variance of projected data while preserving its geometric structure.

**Algorithm Overview and Numerical Stability.** In the following, we highlight further key enhancements of HoroPCA++(HPCA++) from the viewpoint of data representations in the Poincaré Ball. After initialization, the original HoroPCA(HPCA) algorithm iteratively refines candidate ideal points through the following steps: (i) mapping data and ideal points from the Poincaré Ball to the Hyperboloid (ii) computing horospherical projections in the Hyperboloid (iii) mapping projected points back to the Poincaré Ball, and (iv) measuring variance in the low-dimensional representation to update ideal points. This process continues until convergence. HoroPCA++ introduces two critical improvements to this framework. First, we eliminate unnecessary transitions between geometric models, which contribute to computational overhead and accumulate precision errors, by conducting the entire optimization directly in the Hyperboloid. Second, we address fundamental numerical instabilities that arise in horospherical projection computations. Although Chami et al. (2021) correctly identified the computational advantages of the Hyperboloid model – where horospherical projections reduce to solving two linear equation systems – their implementation suffers from ill-conditioned linear systems during optimization, leading to numerical instability. HoroPCA++ resolves this through a novel application of the Sherman–Morrison formula (A.1) combined with regular re-orthonormalization of ideal points, enabling direct computation of matrix inverses for these rank-1 perturbed linear systems and ensuring robust convergence.

| | Balanced Tree | | Phylo Tree | | Diseases | | CS Ph.D. | |
|---|---|---|---|---|---|---|---|---|
| | distortion ($\downarrow$) | $\sigma^2$ ($\uparrow$) | distortion ($\downarrow$) | $\sigma^2$ ($\uparrow$) | distortion ($\downarrow$) | $\sigma^2$ ($\uparrow$) | distortion ($\downarrow$) | $\sigma^2$ ($\uparrow$) |
| BSA | 0.51$\pm$0.00 | 3.01$\pm$0.01 | 0.61$\pm$0.03 | 18.50$\pm$1.76 | 0.56$\pm$0.02 | 4.74$\pm$0.28 | 0.68$\pm$0.04 | 8.07$\pm$0.67 |
| PGA | 0.57$\pm$0.00 | 2.49$\pm$0.01 | 0.61$\pm$0.02 | 16.28$\pm$0.97 | 0.61$\pm$0.03 | 4.76$\pm$0.15 | 0.72$\pm$0.02 | 6.98$\pm$0.67 |
| HPCA | **0.20$\pm$0.00** | **7.15$\pm$0.00** | 0.14$\pm$0.02 | 67.27$\pm$1.74 | 0.15$\pm$0.01 | 15.54$\pm$0.14 | 0.19$\pm$0.03 | **35.48$\pm$1.40** |
| HPCA++ | **0.20$\pm$0.00** | **7.15$\pm$0.00** | **0.12$\pm$0.01** | **68.95$\pm$1.94** | **0.14$\pm$0.00** | **15.64$\pm$0.02** | **0.18$\pm$0.03** | 35.43$\pm$1.33 |

Table 1: Dimensionality reduction performance on 10-dimensional hyperbolic embeddings reduced to two dimensions. Performance is measured by distortion and Fréchet variance ($\sigma^2$). All results reported are averaged over 5 runs. Best in bold.

# 6 EVALUATION

The evaluation of hierarchical clustering is multifaceted. While standard flat clustering metrics such as Normalized Mutual Information (NMI) and the Adjusted Rand Index (ARI) can measure the quality of a specific partition, they are insufficient for assessing the structure of the resulting hierarchy itself. To provide a more comprehensive assessment, we adopt metrics specifically designed for hierarchies. Following prior work in hyperbolic clustering, we use Dasgupta's cost as a primary metric to evaluate the quality and improvement of the learned tree structure. Additionally, we employ Dendrogram Purity, another established metric that measures the homogeneity of clusters throughout the hierarchy. To ensure a robust comparison, we report scores for both of these metrics by constructing dendrograms from our learned embeddings using three standard linkage criteria: average, complete, and ward.

To demonstrate the efficacy of our framework, we conduct a two-part evaluation. First, we assess the quality of the learned embeddings for standard flat clustering tasks against state-of-the-art deep Euclidean methods. Second, we evaluate the quality of the learned hierarchical structure itself, which is the primary goal of our work. All experiments are performed on the MNIST, FMNIST, KMNIST, and GTSRB datasets, with reported results representing the mean and standard deviation across 10 random seeds.

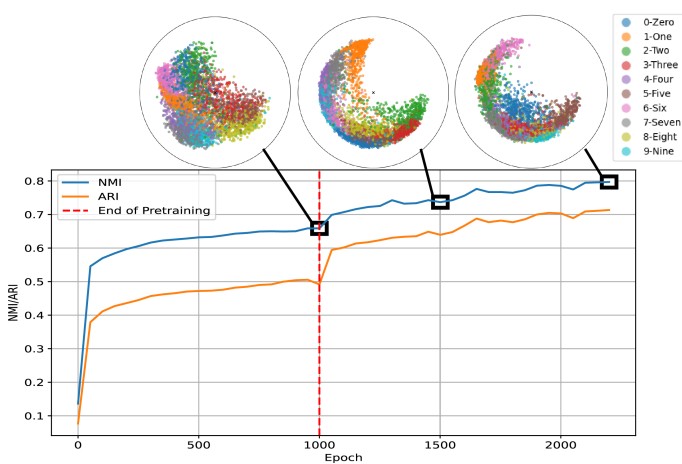

Figure 3: Hyperbolic Embeddings of MNIST on the Poincaré Disk

## 6.1 QUALITY OF LEARNED EMBEDDINGS

We first evaluate the "cluster-friendliness" of the latent space by comparing against prominent deep Euclidean baselines (DEC, IDEC, DCN). Our selection of these foundational methods is intentional. The primary goal of this comparison is not to compete with the latest state-of-the-art in Euclidean flat clustering, but to establish that our hyperbolic autoencoder learns meaningful representations that are competitive with architecturally similar Euclidean models. DEC, IDEC, and DCN are seminal works that, like our method, are based on jointly optimizing an autoencoder's representation with a clustering loss. This allows for a direct and fair comparison of the embedding quality, isolating the geometric differences rather than confounding the results with orthogonal advances from other paradigms like contrastive learning. Table 2 shows the performance of these end-to-end methods alongside our learned embeddings, when paired with standard clustering algorithms such as spectral clustering. Performance is measured using NMI and ARI. Additional results with agglomerative clustering can be found in the appendix Tables 89.

Table 2: Flat Clustering Performance Comparison. Datasets are organized as columns for direct comparison. The highest score for each metric and dataset is in **bold**. Results reported as mean with standard deviation over 10 seeds.

| Method | MNIST | | FMNIST | | KMNIST | | GTSRB | |
|---|---|---|---|---|---|---|---|---|
| | NMI | ARI | NMI | ARI | NMI | ARI | NMI | ARI |
| DEC | 49.6±23.08 | 37.19±20.0 | 55.45±2.0 | 37.52±2.46 | 31.81±3.34 | 19.44±2.70 | 4.19±0.42 | 0.9±0.1 |
| IDEC | 55.53±1.68 | 44.33±1.97 | 56.13±1.29 | 35.72±1.73 | 36.03±1.52 | 22.77±2.06 | 12.96±2.9 | 2.27±0.62 |
| DCN | 55.56±2.24 | 43.77±2.78 | 56.45±1.35 | 36.13±1.87 | 36.16±1.66 | 22.84±2.09 | 13.5±0.39 | 2.64±0.01 |
| Ours (Poincaré) + Spectral Clustering | 72.39±6.3 | 61.38±8.73 | 59.91±1.55 | 41.62±2.84 | 40.01±3.93 | 24.58±3.61 | 33.1±2.16 | 9.±0.677 |
| Ours (Hyperboloid) + Spectral Clustering | **74.3±5.3** | **62.19±9.27** | **62.6±1.32** | **42.57±1.94** | **41.51±3.26** | **24.99±3.59** | **33.22±2.46** | **9.45±0.79** |

## 6.2 EVALUATION OF HIERARCHICAL STRUCTURE

The core contribution of our work is the ability to learn a meaningful hierarchy. We evaluate this directly using two dedicated metrics: Dasgupta's cost, which our model is designed to optimize, and Dendrogram Purity. A lower Dasgupta's cost and a higher Dendrogram Purity signify a higher-quality hierarchy. Additional results can be found in the appendix Table 7.

As shown in Table 3, our method achieves a low Dasgupta's cost, directly validating its success in optimizing the hierarchical objective. Furthermore, the high Dendrogram Purity scores confirm that the resulting hierarchies are composed of homogeneous clusters at all levels. We observe that Ward linkage, which aims to minimize intra-cluster variance, consistently produces the highest quality hierarchies when paired with our embeddings.

Table 3: Hierarchical Quality Assessment. Dasgupta's cost (sampled as described in Section 4.3) and Dendrogram Purity with ward linkage

| Dataset | Metric | Ours (Poincaré) | Ours (Hyperboloid) |
|---------|--------|-----------------|--------------------|
| MNIST | Dasgupta's Cost | $115290 \pm 15395$ | $159634 \pm 20176$ |
|       | Dendrogram Purity | $71.06 \pm 8.78$ | $71.67 \pm 0.1$ |
| FMNIST | Dasgupta's Cost | $34821 \pm 743$ | $166409 \pm 3171$ |
|       | Dendrogram Purity | $47.1 \pm 2.33$ | $51.5 \pm 1.68$ |
| KMNIST | Dasgupta's Cost | $107956 \pm 13785$ | $176307 \pm 5729$ |
|       | Dendrogram Purity | $31.26 \pm 3.36$ | $34.01 \pm 2.74$ |
| GTSRB | Dasgupta's Cost | $83224 \pm 11754$ | $494739 \pm 292744$ |
|       | Dendrogram Purity | $10.4 \pm 0.65$ | $10.9 \pm 1.77$ |

## 6.3 ABLATION STUDY

Since the learning dynamics of deep hierarchical clustering in hyperbolic space are not yet well understood, we conducted several ablation studies to characterize our model's behavior and sensitivity to key design choices. We systematically investigated the following components whose results are reported in the appendix in section A.4.3:

- **Curvature**: Our study on the effect of curvature c on the MNIST dataset revealed that model performance is highly sensitive to this hyperparameter. We found that no single curvature value was optimal across all metrics. For instance, a lower curvature (c=0.1) achieved the best Dendrogram Purity with Ward linkage, while a higher curvature (c=1.0) yielded the best performance with complete and average linkage. This finding underscores that the ideal geometry is task-dependent and supports our conclusion that treating curvature as a fixed hyperparameter is a limitation.

- **Numerical Precision**: We also analyzed the impact of floating-point precision by comparing training runs using 64-bit (Float64) and 32-bit (Float32) precision. Our results showed a notable drop in performance for both NMI and ARI when using the lower 32-bit precision. This confirms that the geometric computations in our model are sensitive to numerical precision and that using 64-bit precision is crucial for achieving stable results.

## 7 CONCLUSION

In this work, we introduced a deep hyperbolic hierarchical clustering framework that addresses critical limitations of prior methods. By presenting a rectified and generalized definition of the hyperbolic Lowest Common Ancestor (LCA), employing a scalable autoencoder architecture, and developing the numerically robust HoroPCA++ for visualization, our approach achieves state-of-the-art performance on several benchmark datasets. The framework is more scalable, geometrically precise, and flexible than previous approaches, supporting both the Poincaré Ball and Hyperboloid models with arbitrary curvature.

Despite these contributions, our work has limitations. The model was evaluated on standard benchmarks of moderate complexity, and its performance on large-scale, high-resolution datasets with a vast number of classes has not been assessed. The current feed-forward network architecture, while effective, may lack the capacity to capture intricate features in more complex data. Furthermore, while our methods improve numerical robustness, computations in hyperbolic space remain sensitive to floating-point precision. Finally, treating the manifold's curvature as a fixed hyperparameter imposes a uniform geometric structure on all datasets, which may not be optimal for capturing their unique intrinsic hierarchies.

Future work will focus on integrating our hyperbolic objective with more powerful network architectures, such as CNNs and Transformers, to tackle more complex datasets. A key direction is to develop methods that allow the manifold's curvature to be learned as a model parameter, enabling the geometry to adapt to the data's intrinsic structure. Scaling these methods to even larger datasets and exploring their application to non-image data, such as text and graphs, remains an exciting avenue for further research.

## REPRODUCIBILITY

Our code and models are publicly available under `https://anonymous.4open.science/r/hyperbolic-clustering-D73C` and our hyperbolic math library `https://anonymous.4open.science/r/hyperbolic-math-1002` as well. Additionally we report all hyperparameters used for our experiments in the appenidx in Section 4.

## LLM USAGE

In some paragraphs, we used LLMs as a post-processing step to improve wording and grammar. While we did not copy anything above sentence level, we drew inspiration for shortening or phrasing more elegantly. Text, figures, and content of the paper are our own work and have **not** been generated, updated, or processed with LLM usage.

## ETHICS STATEMENT

Our work focuses on foundational research in unsupervised learning and hyperbolic geometry, with the potential to advance scientific discovery in various fields. As with any general-purpose clustering algorithm, there is potential for misuse in applications not intended by the authors. This is particularly relevant as we plan to release pretrained models to facilitate reproducibility and future research. We believe the benefits of open research in enabling positive applications outweigh the risks of potential misuse, but we encourage users to consider the ethical implications of their specific applications.

For example, in sensitive domains such as medical diagnostics or financial fraud detection, while our method could help identify meaningful patient subgroups or transaction patterns, algorithmic errors could lead to negative consequences. Such applications must be deployed with caution. Therefore, we strongly recommend that any real-world deployment of this technology, particularly in high-stakes environments, includes robust testing, fairness audits, and meaningful human oversight.

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

## A  APPENDIX

### A.1  MATHEMATICAL PRELIMINARIES

**Fast Inversion of Rank-1 Perturbed Invertible Square Matrices**

**Lemma A.1.** *Let $A \in \mathbb{R}^{n \times n}$ be an invertible square matrix and $\boldsymbol{u}, \boldsymbol{v} \in \mathbb{R}^n$ be column vectors. Then $A + \boldsymbol{u}\boldsymbol{v}^T$ is invertible if and only if $1 + \boldsymbol{v}^T A^{-1} \boldsymbol{u} \neq 0$. In this case, its inverse is given by*

$$(A + \boldsymbol{u}\boldsymbol{v}^\top)^{-1} = A^{-1} - \frac{A^{-1}\boldsymbol{u}\boldsymbol{v}^T A^{-1}}{1 + \boldsymbol{v}^T A^{-1}\boldsymbol{u}}. \tag{5}$$

**Proof**  We refer the reader for this proof to the original paper (Hager, 1989). $\square$

### A.2  HYPERBOLIC LOWEST COMMON ANCESTORS

Before deriving the explicit formulas for geodesic orthogonal projections in the Poincaré Ball and the Hyperboloid, we briefly recap the form of geodesics in each hyperbolic space.

1. **Poincaré Ball:** In the Poincaré Ball, geodesics are arcs of circles that intersect the boundary sphere $\partial \mathbb{P}_c^n$ orthogonally. In the special case where these circles pass through the origin, the geodesics are straight lines through the origin, which occurs when the points are collinear with the origin.

2. **Hyperboloid:** In the Hyperboloid, geodesics are intersections of $\mathbb{H}_c^n$ with 2-dimensional linear subspaces of the ambient Minkowski space $\mathbb{R}^{1,n}$.

**Proposition 4.2**  *Let $\boldsymbol{x}, \boldsymbol{y} \in \mathbb{P}_c^n$ be two points in the Poincaré Ball that are not collinear. The geodesic orthogonal projection of the origin onto the geodesic $\Gamma_{\boldsymbol{x},\boldsymbol{y}}$ connecting $\boldsymbol{x}$ and $\boldsymbol{y}$ is given by:*

$$\boldsymbol{\pi}_\Gamma(\bar{\mathbf{0}}) = \left(1 - \frac{\sqrt{||\boldsymbol{z}||^2 - \frac{1}{c}}}{||\boldsymbol{z}||}\right) \boldsymbol{z}, \quad where \tag{6}$$

$$\boldsymbol{z} = \frac{\left[(1 + c||\boldsymbol{y}||^2)\langle \boldsymbol{x}, \boldsymbol{y}\rangle - (1 + c||\boldsymbol{x}||^2)||\boldsymbol{y}||^2\right]\boldsymbol{x} + \left[(1 + c||\boldsymbol{x}||^2)\langle \boldsymbol{x}, \boldsymbol{y}\rangle - (1 + c||\boldsymbol{y}||^2)||\boldsymbol{x}||^2\right]\boldsymbol{y}}{2\,c\,(|\langle \boldsymbol{x}, \boldsymbol{y}\rangle|^2 - ||\boldsymbol{x}||^2||\boldsymbol{y}||^2)}.$$

**Proof**  By assumption, $\boldsymbol{x}$ and $\boldsymbol{y}$ are non-collinear points on the Poincaré Ball, i.e. the geodesic $\Gamma_{\boldsymbol{x},\boldsymbol{y}}$ is an arc of a circle that meets the boundary $\mathbb{P}_c^n$ orthogonally. Using the circle inversion property with respect to the boundary sphere $\partial \mathbb{P}_c^n = \{\boldsymbol{x} \in \mathbb{R}^n : ||\boldsymbol{x}||^2 = 1/c\}$, we can compute the inverses of $\boldsymbol{x}$ and $\boldsymbol{y}$:

$$\boldsymbol{x}^{-1} = \frac{\boldsymbol{x}}{c||\boldsymbol{x}||^2}, \qquad \boldsymbol{y}^{-1} = \frac{\boldsymbol{y}}{c||\boldsymbol{y}||^2}.$$

These points satisfy $||\boldsymbol{x}|| \cdot ||\boldsymbol{x}^{-1}|| = 1/c$ and $||\boldsymbol{y}|| \cdot ||\boldsymbol{y}^{-1}|| = 1/c$.

**Finding Conditions on the Circle Center**  By construction, the points $\boldsymbol{x}, \boldsymbol{y}, \boldsymbol{x}^{-1}, \boldsymbol{y}^{-1}$ all lie on the same circle $\Gamma$. Thus, we can construct two perpendicular-bisecting hyperplanes to design conditions the center $\boldsymbol{z}$ has to satisfy.

First perpendicular bisector condition:

$$\left\langle \boldsymbol{z} - \frac{\boldsymbol{x} + \boldsymbol{x}^{-1}}{2}, \boldsymbol{x} - \boldsymbol{x}^{-1} \right\rangle = 0 \iff \langle \boldsymbol{z}, \boldsymbol{x}\rangle = \frac{1 + c||\boldsymbol{x}||^2}{2c}. \tag{7}$$

Second perpendicular bisector condition:

$$\left\langle \boldsymbol{z} - \frac{\boldsymbol{y} + \boldsymbol{y}^{-1}}{2}, \boldsymbol{y} - \boldsymbol{y}^{-1} \right\rangle = 0 \iff \langle \boldsymbol{z}, \boldsymbol{y}\rangle = \frac{1 + c||\boldsymbol{y}||^2}{2c}. \tag{8}$$

Span condition:

$$\boldsymbol{z} = \alpha \boldsymbol{x} + \beta \boldsymbol{y}. \tag{9}$$

**Solving for the Coefficients**   Combining condition 7 and 9:

$$\langle \alpha \boldsymbol{x} + \beta \boldsymbol{y}, \boldsymbol{x} \rangle = \alpha ||\boldsymbol{x}||^2 + \beta \langle \boldsymbol{x}, \boldsymbol{y} \rangle = \frac{1 + c||\boldsymbol{x}||^2}{2c} \tag{10}$$

Combining condition 8 and 9:

$$\langle \alpha \boldsymbol{x} + \beta \boldsymbol{y}, \boldsymbol{y} \rangle = \alpha \langle \boldsymbol{x}, \boldsymbol{y} \rangle + \beta ||\boldsymbol{y}||^2 = \frac{1 + c||\boldsymbol{y}||^2}{2c}. \tag{11}$$

Expressing $\alpha$ in equation 10

$$\alpha = \frac{1 + c||\boldsymbol{x}||^2}{2c||\boldsymbol{x}||^2} - \frac{\beta \langle \boldsymbol{x}, \boldsymbol{y} \rangle}{||\boldsymbol{x}||^2} \tag{12}$$

and by substituting for it in condition 11 we can determine $\beta$

$$\beta = \frac{(1 + c||\boldsymbol{x}||^2)\langle \boldsymbol{x}, \boldsymbol{y} \rangle - (1 + c||\boldsymbol{y}||^2)||\boldsymbol{x}||^2}{2\,c\,(|\langle \boldsymbol{x}, \boldsymbol{y} \rangle|^2 - ||\boldsymbol{x}||^2||\boldsymbol{y}||^2)} \tag{13}$$

and $\alpha$ via equation 12

$$\alpha = \frac{(1 + c||\boldsymbol{y}||^2)\langle \boldsymbol{x}, \boldsymbol{y} \rangle - (1 + c||\boldsymbol{x}||^2)||\boldsymbol{y}||^2}{2\,c\,(|\langle \boldsymbol{x}, \boldsymbol{y} \rangle|^2 - ||\boldsymbol{x}||^2||\boldsymbol{y}||^2)}. \tag{14}$$

**Center of the Circle**   The center of the circle $\Gamma$ whose arc is a geodesic connecting $\boldsymbol{x}$ and $\boldsymbol{y}$ is then

$$\boldsymbol{z} = \frac{\left[(1 + c||\boldsymbol{y}||^2)\langle \boldsymbol{x}, \boldsymbol{y} \rangle - (1 + c||\boldsymbol{x}||^2)||\boldsymbol{y}||^2\right] \boldsymbol{x} + \left[(1 + c||\boldsymbol{x}||^2)\langle \boldsymbol{x}, \boldsymbol{y} \rangle - (1 + c||\boldsymbol{y}||^2)||\boldsymbol{x}||^2\right] \boldsymbol{y}}{2\,c\,(|\langle \boldsymbol{x}, \boldsymbol{y} \rangle|^2 - ||\boldsymbol{x}||^2||\boldsymbol{y}||^2)} \tag{15}$$

**Geodesic Projection**   Since the circle $\Gamma$ meets the boundary orthogonally, we have

$$\frac{1}{c} = ||\boldsymbol{z}||^2 - r^2 \quad \Longleftrightarrow \quad r = \sqrt{||\boldsymbol{z}||^2 - \frac{1}{c}} \tag{16}$$

where $r$ is the radius of $\Gamma$. Moreover, we have that the geodesic orthogonal projection $\boldsymbol{\pi}_\Gamma(\bar{\boldsymbol{0}})$ lies on the line from the origin to $\boldsymbol{z}$, and is the point on $\Gamma$ closest to the origin. This gives us:

$$\boldsymbol{\pi}_\Gamma(\bar{\boldsymbol{0}}) = (||\boldsymbol{z}|| - r)\,\frac{\boldsymbol{z}}{||\boldsymbol{z}||} \tag{17}$$

and using 16, we get the formula for the geodesic orthogonal projection

$$\boldsymbol{\pi}_\Gamma(\bar{\boldsymbol{0}}) = (||\boldsymbol{z}|| - r)\,\frac{\boldsymbol{z}}{||\boldsymbol{z}||} = \left(1 - \frac{\sqrt{||\boldsymbol{z}||^2 - \frac{1}{c}}}{||\boldsymbol{z}||}\right) \boldsymbol{z}.$$

$\square$

**Proposition 4.3**   *Let $\boldsymbol{x}, \boldsymbol{y} \in \mathbb{H}^n_c$ be two points in the Hyperboloid. The geodesic orthogonal projection of the origin onto the geodesic $\Gamma_{\boldsymbol{x}, \boldsymbol{y}}$ connecting $\boldsymbol{x}$ and $\boldsymbol{y}$ is given by:*

$$\boldsymbol{\pi}_\Gamma(\bar{\boldsymbol{0}}) = \frac{\alpha \boldsymbol{x} + \beta \boldsymbol{y}}{\sqrt{-c\,\langle \alpha \boldsymbol{x} + \beta \boldsymbol{y}, \alpha \boldsymbol{x} + \beta \boldsymbol{y} \rangle_{\mathcal{L}}}}, \quad where \tag{18}$$

$$\alpha = \frac{\sqrt{c}\,(x_0 + c\langle \boldsymbol{x}, \boldsymbol{y} \rangle_{\mathcal{L}}\,y_0)}{1 - (c\langle \boldsymbol{x}, \boldsymbol{y} \rangle_{\mathcal{L}})^2}, \quad \beta = \frac{\sqrt{c}\,(y_0 + c\langle \boldsymbol{x}, \boldsymbol{y} \rangle_{\mathcal{L}}\,x_0)}{1 - (c\langle \boldsymbol{x}, \boldsymbol{y} \rangle_{\mathcal{L}})^2}.$$

**Proof** The geodesic orthogonal projection of $\pi_{\Gamma}(\bar{\mathbf{0}})$ can be computed in two steps:

1. Compute the orthogonal projection $\pi_{\text{span}\{x,y\}}(\bar{\mathbf{0}})$ of the origin $\bar{\mathbf{0}}$ onto the linear subspace $\text{span}\{x, y\}$ with respect to the Minkwoski inner product $\langle ., . \rangle_{\mathcal{L}}$.

2. Rescale $\pi_{\text{span}\{x,y\}}(\bar{\mathbf{0}})$ to be a point on the Hyperboloid:
$$\pi_{\Gamma}(\bar{\mathbf{0}}) = \frac{\pi_{\text{span}\{x,y\}}(\bar{\mathbf{0}})}{\sqrt{-c \left\langle \pi_{\text{span}\{x,y\}}(\bar{\mathbf{0}}), \pi_{\text{span}\{x,y\}}(\bar{\mathbf{0}}) \right\rangle_{\mathcal{L}}}}$$

Let $M = [x|y] \in \mathbb{R}^{(n+1)\times 2}$ denote the matrix with column-vectors $x, y \in \mathbb{R}^{n+1}$, $\bar{\mathbf{0}} \in \mathbb{R}^{n+1}$ the Hyperboloid's origin written as column-vector, $B = \text{diag}[-1, 1, \ldots, 1] \in \mathbb{R}^{(n+1)\times(n+1)}$ the matrix associated with the Minkowski inner product, and $\rho := \langle x, y \rangle_{\mathcal{L}}$. The orthogonal projection $\pi_{\text{span}\{x,y\}}(\bar{\mathbf{0}})$ with respect to $\langle ., . \rangle_{\mathcal{L}}$ is given by

$$\pi_{\text{span}\{x,y\}}(\bar{\mathbf{0}}) = M(M^T B M)^{-1} M^T B \bar{\mathbf{0}}$$

Since $M^T B M \in \mathbb{R}^{2\times 2}$ is a symmetric matrix with main diagonal entries $\{\langle x, x \rangle_{\mathcal{L}}, \langle y, y \rangle_{\mathcal{L}}\} = \{-1/c, -1/c\}$ and off-diagonal element $\rho$ we can apply Lemma A.1 to directly express the inverse. For this we set $A = (-1/c - \rho)\,\mathbb{I}_2 \in \mathbb{R}^{2\times 2}$, $u = (\rho, \rho)^T \in \mathbb{R}^2$, and $v = (1, 1) \in \mathbb{R}^2$. Then $A$ is invertible and $1 + v^T A^{-1} u = \frac{1-c\rho}{1+c\rho} \neq 0$ since $x \neq y$. Thus,

$$(M^T B M)^{-1} = (A + uv^\top)^{-1} = A^{-1} - \frac{A^{-1}uv^T A^{-1}}{1 + v^T A^{-1}u} = \frac{-c}{1-(c\rho)^2}\begin{bmatrix} 1 & c\rho \\ c\rho & 1 \end{bmatrix}$$

Next, we compute $M^T B \bar{\mathbf{0}} = (-x_0/\sqrt{c}, -y_0/\sqrt{c})^T \in \mathbb{R}^2$ and put everything together:

$$\pi_{\text{span}\{x,y\}}(\bar{\mathbf{0}}) = M(M^T B M)^{-1}M^T B \bar{\mathbf{0}} = [x|y]\frac{-c}{1-(c\rho)^2}\begin{bmatrix} 1 & c\rho \\ c\rho & 1 \end{bmatrix}\begin{bmatrix} -\frac{x_0}{\sqrt{c}} \\ -\frac{y_0}{\sqrt{c}} \end{bmatrix} =$$

$$= \frac{\sqrt{c}}{1-(c\rho)^2}\left((x_0 + c\rho y_0)x + (c\rho x_0 + y_0)y\right).$$

Re-substituting for $\rho$ and defining $\alpha = \frac{\sqrt{c}\,(x_0 + c\langle x,y \rangle_{\mathcal{L}}\,y_0)}{1-(c\langle x,y \rangle_{\mathcal{L}})^2}$, $\beta = \frac{\sqrt{c}\,(y_0 + c\langle x,y \rangle_{\mathcal{L}}\,x_0)}{1-(c\langle x,y \rangle_{\mathcal{L}})^2}$ yields $\pi_{\text{span}\{x,y\}}(\bar{\mathbf{0}}) = \alpha x + \beta y$. Finally, rescaling $\pi_{\text{span}\{x,y\}}(\bar{\mathbf{0}})$ such that it is a point on the Hyperboloid yields the desired result. $\qquad\square$

### A.3 HOROPCA++

To ensure that the hyperbolic PCA components reflect the true variance and relationships within the data, we frechét mean center the data prior to employing HoroPCA and HoroPCA++. For ease of exposition, we present a side-by-side comparison of the original HoroPCA and our improved HoroPCA++ for inputs $x \in \mathbb{P}_c^n$ and $Q = \left[q_1^T | \ldots | q_k^T\right]$ for $q_i \in \mathcal{I}_{\mathbb{P}}^{n-1}$.

**HoroPCA**

0. –

1. Repeat:
   (a) $Q_{\mathbb{P}}^{\perp} = QR[Q_{\mathbb{P}}]$
   (b) $Q_{\mathbb{H}} = \phi_{\mathcal{I}}(Q_{\mathbb{P}}^{\perp})$
   (c) $x_{\mathbb{H}} = \phi(x_{\mathbb{P}})$
   (d) # Horospherical Projection
       Compute $\pi_{Q_{\mathbb{H}}}(x_{\mathbb{H}})$
   (e) $\pi_{Q_{\mathbb{P}}}(x_{\mathbb{P}}) = \psi(\pi_{Q_{\mathbb{H}}}(x_{\mathbb{H}}))$
   (f) Compute $\mathcal{L}_{\mathbb{P}}(\pi_{Q_{\mathbb{P}}}(x_{\mathbb{P}}))$
   (g) Update $Q_{\mathbb{P}}$

**HoroPCA++**

0. $x_{\mathbb{H}} = \phi(x_{\mathbb{P}})$

1. Repeat:
   (a) $Q_{\mathbb{P}}^{\perp} = QR[Q_{\mathbb{P}}]$
   (b) $Q_{\mathbb{H}} = \phi_{\mathcal{I}}(Q_{\mathbb{P}}^{\perp})$
   (c) –
   (d) # Horospherical Projection
       Compute $\pi_{Q_{\mathbb{H}}}(x_{\mathbb{H}})$
   (e) –
   (f) Compute $\mathcal{L}_{\mathbb{H}}(\pi_{Q_{\mathbb{H}}}(x_{\mathbb{H}}))$
   (g) Update $Q_{\mathbb{P}}$

Here, $Q_{\mathbb{P}}^{\perp}$ denotes the row-wise orthogonalized matrix of ideal points, $\phi, \phi_{\mathcal{L}}, \psi$ the mappings between them with respect to the hyperbolic space models, $\mathcal{L}_{\mathbb{P}}, and \mathcal{L}_{\mathbb{H}}$ the loss functions measuring the generalized variance $-var = -(d_{\mathcal{M}}^{pw}(\pi_{\mathcal{M}}))^2/n$ with $d_{\mathcal{M}}^{pw}(\pi_{\mathcal{M}})$ denoting the pair-wise geodesic distance

of the projected points $\pi_{\mathcal{M}}$. In particular, for both algorithms to work we need to define mappings that maps ideal points from the Poincare Ball to the Hyperboloid and vice-versa. The following mappings are generalized to arbitrary curvatures and are compatible with the usual stereographic projections (Chami et al., 2021; Mishne et al., 2023).

**Definition A.2.** Let $x \in \mathcal{I}_{\mathbb{P}}^{n-1}$ and $y \in \mathcal{I}_{\mathbb{H}}^{n-1}$. The maps

$$\phi_{\mathcal{I}} : \mathcal{I}_{\mathbb{P}}^{n-1} \to \mathcal{I}_{\mathbb{H}}^{n-1}, \quad x \mapsto \left(1, \sqrt{c}\, x_1, \ldots, \sqrt{c}\, x_n\right), \tag{19}$$

$$\psi_{\mathcal{I}} : \mathcal{I}_{\mathbb{H}}^{n-1} \to \mathcal{I}_{\mathbb{P}}^{n-1}, \quad y \mapsto \left(\frac{y_1}{\sqrt{c}\, y_0}, \ldots, \frac{y_n}{\sqrt{c}\, y_0}\right) \tag{20}$$

**Horospherical Projection** To compute the horospherical projection in the Hyperboloid the following two linear equation systems need to be solved to compute the coefficients $z$ of data $x$ being projected onto linear Minkowski subspaces: (i) $Q_{\mathbb{H}}^{\perp} B (Q_{\mathbb{H}}^{\perp})^T z = Q_{\mathbb{H}}^{\perp} B x$ (ii) $Q_{\mathbb{H}}^{\perp} B (Q_{\mathbb{H}}^{\perp})^T z = Q_{\mathbb{H}}^{\perp} B x$. Here, $B = \text{diag}[-1, 1, \ldots, 1] \in \mathbb{R}^{(n+1) \times (n+1)}$ is the matrix associated with the Minkowski inner product. Both systems can be efficiently solved by computing the inverse $(Q_{\mathbb{H}}^{\perp} B (Q_{\mathbb{H}}^{\perp})^T)^{-1}$. Through the orthogonalization of $Q_{\mathbb{P}}$ in the algorithm and the mapping $\phi_{\mathcal{I}}$ we ensure that the lower $(n \times n)$ submatrix of $Q_{\mathbb{H}}^{\perp} B (Q_{\mathbb{H}}^{\perp})^T$ is orthonormal. Further, since the rows of $Q_{\mathbb{H}}$ are vectors of the asymptotic null cone $\mathcal{I}_{\mathbb{H}}^{n-1}$ both the first row and column of $Q_{\mathbb{H}}^{\perp} B (Q_{\mathbb{H}}^{\perp})^T$ is the unit vector $[1, \mathbf{0}]^T \in \mathbb{R}^{n+1}$. This allows us to describe $Q_{\mathbb{H}}^{\perp} B (Q_{\mathbb{H}}^{\perp})^T$ as rank-1 perturbed matrix and compute its inverse directly via the Sherman-Morrison formula (Lemma A.1) making the solution robust to numerical imprecisions. For this we set $A = \mathbb{I}_n \in \mathbb{R}^{(n+1) \times (n+1)}$, $u = (-1, \ldots, -1)^T \in \mathbb{R}^{n+1}$, and $v = (1, \ldots, 1) \in \mathbb{R}^{n+1}$.

**HoroPCA Experiments** To compare HoroPCA++ against its predecessor HoroPCA, we preprocess all datasets in accordance with the original paper and average the results over 5 runs using seed 44. We compare the differences on the task of dimensionality reduction which is of our interest. More specifically, we report on all datasets which includes a fully balanced tree, a phylogenetic tree, a biological graph comprising of diseases' relationships and a graph of Computer Science (CS) Ph.D. advisor-advisee relationships (Sala et al., 2018). These datasets have 40, 344, 516 and 1025 nodes, respectively. As evaluation metrics we compute the distance-preservation after projection $\pi$, measured by the average distortion

$$\text{distortion} = \frac{1}{\binom{|S|}{2}} \sum_{x \neq y \in S} \frac{|d_{\mathcal{M}}(\pi(x), \pi(y)) - d_{\mathcal{M}}(x, y)|}{d_{\mathcal{M}}(x, y)}$$

and the frechét variance

$$\sigma_{\mathcal{M}}^2(S) := \frac{1}{|S|} \sum_{x \in S} d_{\mathcal{M}}(x, \mu(S))^2,$$

where $\mu(S)$ is frechét mean.

## A.4 Experiments

### A.4.1 Hyperbolic Layers

In our architecture we employ two distinct hyperbolic linear encoding layers (Figure 1), each tailored to a specific model of hyperbolic space:

**Poincaré Ball Linear Layer++ (HyperbolicLinearPP)** Shimizu et al. (2021) hyperbolic linear layer generalizes the Euclidean fully-connected layer to the Poincaré Ball model by interpreting the output coordinates of the affine transformation $y_k = \langle a_k, x \rangle - b_k$ as signed hyperbolic distances to the Euclidean coordinate axes:

$$\boldsymbol{y} = f^{\boldsymbol{z}, r}(\boldsymbol{x}) = \frac{\boldsymbol{w}}{1 + \sqrt{1 + c\|\boldsymbol{w}\|^2}}, \quad \text{where } \boldsymbol{w} = \left(\frac{1}{\sqrt{c}} \sinh\left(\sqrt{c}\, v_k^{\boldsymbol{z}, r}(\boldsymbol{x})\right)\right)_{k=1}^K, \tag{21}$$

$$v_k^{\boldsymbol{z}, r}(\boldsymbol{x}) = \frac{2\|\boldsymbol{z}\|}{\sqrt{c}} \sinh^{-1}\left((1 - \lambda_x) \sinh(2\sqrt{c}\, r) + \sqrt{c}\, \lambda_{\boldsymbol{x}} \cosh(2\sqrt{c}\, r) \left\langle \frac{\boldsymbol{z}}{\|\boldsymbol{z}\|}, \boldsymbol{x} \right\rangle\right). \tag{22}$$

**Fully Hyperbolic Convolutional Neural Networks (FHCNN)** Bdeir et al. (2024); Chen et al. (2021) hyperbolic linear layer generalizes the Euclidean fully-connected layer for the Hyperboloid model by first applying a linear operation ($W\boldsymbol{x} + \boldsymbol{b}$) to the hyperbolic input $x \in \mathbb{H}_c^n$ followed by a normalization ensuring that the layer's output lies on the Hyperboloid and falls within a certain range:

$$\boldsymbol{y} = f^{W,\boldsymbol{b}}(\boldsymbol{x}) = \begin{bmatrix} \sqrt{\left(\lambda\sigma(\alpha_0^{W,\boldsymbol{b}})\right)^2 + \frac{1}{c}} \\ \lambda\sigma(\alpha_0)\frac{\boldsymbol{\alpha}_{1:n}}{||\boldsymbol{\alpha}_{1:n}||} \end{bmatrix} \quad \text{where} \quad \boldsymbol{\alpha}^{W,\boldsymbol{b}} = (\alpha_0^{W,\boldsymbol{b}}, \boldsymbol{\alpha}_{1:n}^{W,\boldsymbol{b}})^T = W\boldsymbol{x} + \boldsymbol{b}, \quad (23)$$

where $\sigma$ is the sigmoid function, $\boldsymbol{b}$ is the bias term, and $\lambda > 0$ controls the layer's range.

### A.4.2 HYPERPARAMETER

| Parameter | Value |
|---|---|
| *General Training* | |
| Seeds | 728, 395, 291, 18, 887, 93, 108, 480, 868, 134 |
| Batch size | 512 |
| Optimizer | RiemannianSGD |
| Precision | float64 |
| *Pretraining* | |
| Epochs | 1000 |
| Learning rate | $1 \times 10^{-3}$ |
| Weight decay | $1 \times 10^{-4}$ |
| *Clustering* | |
| Epochs | 1200 |
| Learning rate | $1 \times 10^{-2}$ |
| Weight decay | $1 \times 10^{-4}$ |
| LR Scheduler | Warmup Cosine (20 epochs) |
| *Architecture* | |
| Model Type | Feed-forward AE |
| Hidden Dims | [512, 256, 64, 16] |
| Latent Dim | 4 (MNIST/KMNIST), 6 (FMNIST) |
| Manifold | Poincaré Ball |
| Curvature ($c$) | 0.1 |
| Activation | Swish |
| Hyperbolic Encoding | HyperbolicLinearPP(Poincare)/FHCNN(Hyperboloid) |
| Hyperbolic Decoding | Log_EuclideanLinear |
| *LCA Loss* | |
| Temperature ($\tau$) | 0.01 |
| Subsample Size | 150 |

Table 4: Hyperparameters for all dataset experiments(MNIST, FMNIST, KMNIST, GTSRB).

### A.4.3 ABLATION STUDIES RESULTS

Table 5 shows the performance on MNIST across Dendrogram Purity (DP), Spectral NMI, and ARI for different values of curvature c. The following results were obtained on MNIST using the same seed and spectral clustering. In Table 6 we compare the float 64-bit precision to float 32-bit. The performance drop with 32-bit precision suggests that some geometric operations, like orthogonal projections or distance calculations, accumulate errors that can degrade the quality of the learned embeddings.

Table 5: Ablation study on the effect of curvature $c$ for the MNIST dataset. All other hyperparameters are held constant as described in Table 4. Performance is measured by Dendrogram Purity (DP, Ward and Complete linkage), Spectral Clustering NMI, and Adjusted Rand Index (ARI). The best performance for each metric is highlighted in **bold**.

| Curvature ($c$) | DP Ward | DP Complete | Spectral NMI | ARI |
|---|---|---|---|---|
| 10.0 | 64.6 | 46.03 | 68.1 | 56.9 |
| 7.5 | 55.4 | 42.6 | 58.1 | 43.1 |
| 5.0 | 62.9 | 50.86 | 74.2 | 65.3 |
| 2.5 | 76.7 | 55.42 | 78.0 | 70.2 |
| 2.0 | 73.5 | 56.27 | 76.6 | 69.6 |
| 1.0 | 76.9 | **65.0** | 69.82 | 69.8 |
| 0.5 | 72.8 | 57.93 | 74.1 | 62.9 |
| 0.1 | **81.3** | 64.6 | **79.7** | **71.4** |

Table 6: Comparison of numerical precision on MNIST for spectral clustering. The Float32 results represent the mean and standard deviation over 3 seeds.

| Model | Precision | NMI Score (%) |
|---|---|---|
| Poincaré Ball | Float32 | $74.22 \pm 1.8$ |
| | Float64 | $78.96 \pm 0.642$ |
| Hyperboloid | Float32 | $75.69 \pm 4.6$ |
| | Float64 | $77.81 \pm 0.025$ |

## A.5 ADDITIONAL RESULTS

In this section we provide additional results for dendrogram purity and agglomerative clustering.

Table 7: Dendrogram Purity scores across different datasets, models, and linkage methods. Values are reported as mean $\pm$ standard deviation.

| Dataset | Model | Complete | Average | Single | Ward |
|---|---|---|---|---|---|
| FMNIST | Poincaré | $0.393 \pm 0.030$ | $0.464 \pm 0.020$ | $0.363 \pm 0.026$ | $0.471 \pm 0.023$ |
| | Hyperboloid | $0.448 \pm 0.007$ | $0.501 \pm 0.018$ | $0.391 \pm 0.022$ | $0.515 \pm 0.019$ |
| GTSRB | Poincaré | $0.096 \pm 0.004$ | $0.110 \pm 0.007$ | $0.090 \pm 0.002$ | $0.104 \pm 0.006$ |
| | Hyperboloid | $0.089 \pm 0.010$ | $0.105 \pm 0.020$ | $0.084 \pm 0.019$ | $0.109 \pm 0.018$ |
| KMNIST | Poincaré | $0.268 \pm 0.029$ | $0.315 \pm 0.034$ | $0.208 \pm 0.028$ | $0.313 \pm 0.035$ |
| | Hyperboloid | $0.280 \pm 0.013$ | $0.328 \pm 0.020$ | $0.238 \pm 0.021$ | $0.34 \pm 0.027$ |
| MNIST | Poincaré | $0.551 \pm 0.087$ | $0.713 \pm 0.069$ | $0.538 \pm 0.073$ | $0.710 \pm 0.088$ |
| | Hyperboloid | $0.562 \pm 0.073$ | $0.718 \pm 0.066$ | $0.549 \pm 0.047$ | $0.716 \pm 0.100$ |

Table 8: Agglomerative Clustering NMI scores for different models and linkage methods. Values are reported as mean $\pm$ standard deviation.

| Dataset | Model | Complete | Average | Single |
|---------|-------|----------|---------|--------|
| FMNIST | Poincaré | $0.498 \pm 0.029$ | $0.567 \pm 0.031$ | $0.002 \pm 0.0001$ |
|  | Hyperboloid | $0.537 \pm 0.013$ | $0.593 \pm 0.011$ | $0.002 \pm 0.0001$ |
| GTSRB | Poincaré | $0.300 \pm 0.010$ | $0.324 \pm 0.017$ | $0.013 \pm 0.004$ |
|  | Hyperboloid | $0.218 \pm 0.091$ | $0.221 \pm 0.108$ | $0.031 \pm 0.018$ |
| KMNIST | Poincaré | $0.313 \pm 0.035$ | $0.341 \pm 0.033$ | $0.002 \pm 0.0005$ |
|  | Hyperboloid | $0.304 \pm 0.021$ | $0.347 \pm 0.027$ | $0.002 \pm 0.0002$ |
| MNIST | Poincaré | $0.599 \pm 0.079$ | $0.717 \pm 0.045$ | $0.002 \pm 0.0003$ |
|  | Hyperboloid | $0.606 \pm 0.063$ | $0.722 \pm 0.032$ | $0.002 \pm 0.0002$ |

Table 9: Agglomerative Clustering ARI scores for different models and linkage methods. Values are reported as mean $\pm$ standard deviation.

| Dataset | Model | Complete | Average | Single |
|---------|-------|----------|---------|--------|
| FMNIST | Poincaré | $0.325 \pm 0.034$ | $0.383 \pm 0.037$ | $\approx 0.0$ |
|  | Hyperboloid | $0.368 \pm 0.014$ | $0.401 \pm 0.021$ | $\approx 0.0$ |
| GTSRB | Poincaré | $0.076 \pm 0.007$ | $0.086 \pm 0.005$ | $0.0001 \pm 0.0001$ |
|  | Hyperboloid | $0.035 \pm 0.040$ | $0.042 \pm 0.050$ | $0.0011 \pm 0.0012$ |
| KMNIST | Poincaré | $0.174 \pm 0.030$ | $0.180 \pm 0.025$ | $\approx 0.0$ |
|  | Hyperboloid | $0.158 \pm 0.017$ | $0.177 \pm 0.017$ | $\approx 0.0$ |
| MNIST | Poincaré | $0.447 \pm 0.096$ | $0.564 \pm 0.077$ | $\approx 0.0$ |
|  | Hyperboloid | $0.455 \pm 0.086$ | $0.575 \pm 0.046$ | $\approx 0.0$ |

A.6  Cost Function Formulation

**Wang's Equivalent Formulation**  Dasgupta's 3.2 original formulation's use of $|\text{leaves}(T[i \vee j])|$ makes it difficult to optimize directly. Thus, Wang (Wang & Wang, 2018) restated $C_{\text{Dasgupta}}(T; w)$ in terms of the relationships between all triplets $(i, j, k)$:

$$C_{\text{Wang}}(T; w) = \sum_{i,j,k} (w_{ij} + w_{ik} + w_{jk} - w_{ijk}(T; w)) + 2 \sum_{i,j} w_{ij}, \tag{24}$$

where $w_{ijk}(T; w) = w_{ij}\mathbb{1}[\{i,j\}|k] + w_{ik}\mathbb{1}[\{i,k\}|j] + w_{jk}\mathbb{1}[\{j,k\}|i]$ and $\mathbb{1}[\{i,j\}|k] = 1$ if "$i \vee j$ is a descendant of $i \vee j \vee k$".

## A.7 HYPERBOLIC ENCODING AND DECODING MAPS

The following table details the formulas for the exponential ($\exp_0$) and logarithmic ($\log_0$) maps used for hyperbolic encoding and decoding.

| | Poincaré Ball | Hyperboloid |
|---|---|---|
| **Origin** | $\bar{\mathbf{0}} = (0, 0, \dots, 0)$ | $\bar{\mathbf{0}} = (1/\sqrt{c}, 0, \dots, 0)$ |
| $\mathbf{Exp}_{\bar{\mathbf{0}}}(\boldsymbol{v})$ | $\boldsymbol{v} \mapsto \dfrac{\tanh\left(\sqrt{c}\,\|\boldsymbol{v}\|\right)}{\sqrt{c}\,\|\boldsymbol{v}\|}\,\boldsymbol{v}$ | $\boldsymbol{v} \mapsto \cosh\left(\sqrt{c\,\langle\boldsymbol{v},\boldsymbol{v}\rangle_{\mathcal{L}}}\right)\bar{\mathbf{0}} + \dfrac{\sinh\left(\sqrt{c\,\langle\boldsymbol{v},\boldsymbol{v}\rangle_{\mathcal{L}}}\right)}{\sqrt{c\,\langle\boldsymbol{v},\boldsymbol{v}\rangle_{\mathcal{L}}}}\,\boldsymbol{v}$ |
| $\mathbf{Log}_{\bar{\mathbf{0}}}(\boldsymbol{x})$ | $\boldsymbol{x} \mapsto \dfrac{\tanh^{-1}\left(\sqrt{c}\,\|\boldsymbol{x}\|\right)}{\sqrt{c}\,\|\boldsymbol{x}\|}\,\boldsymbol{x}$ | $\boldsymbol{x} \mapsto \left(0, \dfrac{\cosh^{-1}(\sqrt{c}\,x_0)}{\sqrt{c\,x_0^2 - 1}}\,\boldsymbol{x}\right)$ |
| $d_{\bar{\mathbf{0}}}(\boldsymbol{x})$ | $\dfrac{2\tanh^{-1}\left(\sqrt{c}\,\|\boldsymbol{x}\|\right)}{\sqrt{c}}$ | $\dfrac{\cosh^{-1}\left(\sqrt{c}\,x_0\right)}{\sqrt{c}}$ |

Table 10: Summary of encoding/decoding operations in the Poincaré Ball and the Hyperboloid model

