# OpenReview forum: "Deep Hyperbolic Hierarchical Clustering"
_ICLR.cc/2026/Conference — ICLR 2026 Conference Withdrawn Submission_

### Official Review · Reviewer_YrYM · 2025-10-27

**Soundness:** 2
**Presentation:** 3
**Contribution:** 2
**Rating:** 6
**Confidence:** 4

**Summary:**

The paper proposes a deep framework for hierarchical clustering in hyperbolic space that  rectifies and generalizes the definition of the hyperbolic lowest common ancestor (LCA) for both the Poincaré ball and the hyperboloid, adds scalability via an autoencoder that learns cluster-friendly hyperbolic latents with a differentiable relaxation of Dasgupta’s cost, and (iii) introduces HoroPCA++, a numerically robust dimensionality reduction method that avoids model-switching and stabilizes horospherical projections.

**Strengths:**

The paper pinpoints a flaw in the prior hyperbolic LCA  and provides a rectified LCA that behaves consistently with tree LCAs across Poincaré and hyperboloid models. This addresses a concrete gap in Chami et al. (2020) and is well motivated. Also, supporting both Poincaré and hyperboloid formalisms increases applicability and makes the geometric claims more model agnostic.

**Weaknesses:**

The paper’s empirical scope is narrow, experiments are limited to moderate-scale image datasets (MNIST, FMNIST, KMNIST, GTSRB), so claims of improved scalability are not stress tested on larger or non image settings such as graphs or text. Baselines focus on classic Euclidean deep clustering methods, which while contextualizing the approach, do not establish competitiveness against stronger modern self supervised clustering pipelines.

**Questions:**

It is possible to extend your results (both theoretical and empirical) to isometric models? e.g. Klein model or upper half plane.

---

### Official Review · Reviewer_azfU · 2025-10-27

**Soundness:** 2
**Presentation:** 1
**Contribution:** 2
**Rating:** 2
**Confidence:** 3

**Summary:**

This paper introduces a hyperbolic clustering where input data is first encoded into a low dimensional hyperbolic latent space using some autoencoder and then clustered using a slight adaptation of (Chami et al., 2020)'s hyperbolic hierarchical clustering method. Moreover, the authors present a more efficient and stable version of HoroPCA. Both of these methods have also been adapted to allow different choices of negative curvatures.

**Strengths:**

- Most of the text is quite well written, particularly the first half, making it quite easy to read through.
- The proposed HoroPCA++ seems like it could be quite interesting, although I'm not certain due to some details missing.
- It's nice that the authors have included a variable curvature into the existing HoroPCA and hyperbolic clustering formulations.

**Weaknesses:**

The most glaring weakness in my opinion is the fact that the authors have added far too little references, both to other works as well as to parts included in their own paper. Here is an incomplete list of some examples:
- In the "hyperbolic representation learning" section of the related work the authors have included almost no recent works, only discussing some older theoretical hyperbolic works, aside from (Sala et al., 2018). This section should contain far more discussion on the many recent hyperbolic representation learning papers, especially given the claims in lines 83-86. See for example the surveys [1, 2] as a starting point.
- The background on hyperbolic space (Subsection 3.1) has zero references. Even appendix A.7, containing some of the formulas, is presented without a single reference.
- Horospheres (Definition 5.1) are presented without a reference.
- The Section on HoroPCA++ (5), contains no reference to the Appendix where all the details are presented.
- This same section also contains Table 1, which is presented without details and without any reference to this table whatsoever.
- At the start of the evaluation section (6), the authors claim to follow prior work (line 367), but do not include any reference to any of this prior work.
- Line 369 mentions "Dendogram Purity" as an established metric, but again without a reference.

There are many more examples to be found throughout the paper and this lack of referencing has several negative consequences. For example, it is quite difficult to establish what the authors claim as their own work versus what they have taken from existing work. Moreover, as someone less familiar with unsupervised clustering, I am not certain what some of the incorporated metrics are. Without proper referencing, it is difficult to review this paper with much confidence.

Regarding the contents of the paper, I think there are some additional weaknesses:
1.  The novelty of the clustering method seems limited. Unless I am mistaken, the authors have taken the method from (Chami et al., 2020) and have added a variable curvature and a small fix to the hyperbolic LCA formulation. The proof for the variable curvature seems like a straightforward adaptation of the original proof by (Chami et al., 2020). For the hyperbolic LCA fix, I'm not sure how much effect it actually has in practice and this is not tested in the experiments. The paper should contain some analysis into how often the problem with (Chami et al., 2020)'s formulation actually arises in practice and how much of a difference their improvement makes, as I suspect that it does not lead to a significant difference.
2. The HoroPCA++ part seems interesting, but sadly all of the interesting details have been moved to the appendix (without reference), where these have been presented too concisely to judge the soundness of their formulation. I think it would be better to address this part properly in the main paper.
3. I also cannot judge the experiments regarding HoroPCA++ since there are no details included (also not enough details in Appendix A.3). How are these methods applied to trees?
4. Maybe I've missed it, but I could not find a description of the encoder that is used to generate low dimensional features (before moving to hyperbolic space). More explicitly, I'm unsure about the architecture of the "E.Encoder" (and also "E.Decoder") block in Figure 1.
5. As mentioned by the authors, the clustering method is compared against fairly old baselines on small datasets. I think the paper should at least include some larger datasets. I am not too familiar with the clustering literature and the related work section only contains the 3 baselines as references, so I cannot say much about whether the baselines are well justified. However, I would like to ask: why can the method not be compared properly to more recent baselines?

Some small additional notes:
1. The fraction in the definition of the Riemannian metric of the Poincaré ball (line 140) should be squared. Same in the set of inner products in lines 142-143.
2. I think the first case in Definition 4.1 can be dropped, since it is contained in the last case. A projection is mathematically defined as an idempotent mapping, so when a point is already on a geodesic, its orthogonal projection onto it is the identity.


[1] Peng, Wei, et al. "Hyperbolic deep neural networks: A survey." IEEE Transactions on pattern analysis and machine intelligence 44.12 (2021): 10023-10044.
[2]Mettes, Pascal, et al. "Hyperbolic deep learning in computer vision: A survey." International Journal of Computer Vision 132.9 (2024): 3484-3508.

**Questions:**

See under weaknesses.

---

### Official Review · Reviewer_4T2Y · 2025-10-29

**Soundness:** 2
**Presentation:** 1
**Contribution:** 1
**Rating:** 2
**Confidence:** 4

**Summary:**

The paper presents: (1) an autoencoder based on reconstruction loss and hyperbolic lowest common ancestor (LCA) loss, and (2) a hyperbolic dimensional reduction method HoroPCA++.

**Strengths:**

The authors explore the hyperbolic tool in machine learning tasks.

**Weaknesses:**

1. The paper presents itself as a contribution to hierarchical clustering. It argues that the hyperbolic lowest common ancestor (LCA) of Chami et al. (2020) is not precisely defined, motivating Definition 4.1 and an autoencoder trained with the continuous Dasgupta cost together with the reconstruction loss. However, the method does not produce a hierarchy or a clustering algorithm. The stated positioning is misleading.
2. The paper reads as a reply to two specific papers from Chami et al rather than a clear positioning within hyperbolic hierarchical clustering, hyperbolic dimensionality reduction, hyperbolic representation learning, and hyperbolic metric learning. A broader map of the area is missing. Why are autoencoder + LCA and HoroPCA++ interesting?
3. The transition from deep hyperbolic hierarchical clustering to hyperbolic dimensionality reduction is abrupt. What is the conceptual link, and how do Sections 4–5 connect?
4. The “wrong LCA point” observation in Figure 2 is not an issue for hierarchical clustering methods such as HYPHC; the goal of hierarchical clustering is to build a hierarchy where points are leaves. In that setting, a single data point does not become the ancestor of another single point. Do the authors observe such issues for hierarchical clustering tasks?
5. A very large body of related work are not included and not compared to. The paper would need a thorough literature review and empirical comparison, specific to hyperbolic representation learning, hyperbolic dimensionality reduction, hyperbolic hierarchical clustering, and hyperbolic metric learning. Especially for the tasks that are related to what the authors considered: Unsupervised Hyperbolic Metric Learning, MHCN: A Hyperbolic Neural Network Model for Multi-view Hierarchical Clustering, CO-SNE: Dimensionality Reduction and Visualization for Hyperbolic Data, Nested Hyperbolic Spaces for Dimensionality Reduction and Hyperbolic NN Design, Representation Tradeoffs for Hyperbolic Embeddings, From trees to continuous embeddings and back: Hyperbolic hierarchical clustering, Gradient-based hierarchical clustering using continuous representations of trees in hyperbolic space. For hyperbolic LCA:  Tree-Wasserstein Distance for High Dimensional Data with a Latent Feature Hierarchy. Recent work in hyperbolic representation learning more broadly: Poincaré embeddings for learning hierarchical representations, Learning continuous hierarchies in the Lorentz model of hyperbolic geometry, Neural embeddings of graphs in hyperbolic space, Hyperbolic representation learning: Revisiting and advancing, to name but a few.
6. Definition 4.1 is introduced without an illustrative figure and thus makes it somewhat hard to follow. For better illustration, could the authors add a diagram that shows the geodesic, the projection point, angles, and how $\\cos(\\cdot)$ is computed in both models?
7. The notation $\\pi_{\\Gamma}(\bar{0})$ is confusing. The projection depends on $\(x, y\)$, but this dependence does not appear in the symbol. The dependence should be explicit.
8. Propositions 4.2–4.3: the “geodesic orthogonal projection of the origin” is not properly defined before deriving a closed form. Could the authors explain the object first, then present the formula as a property? Also, what are the implications of propositions 4.2 and 4.3? What is the message that the authors want to convey?
9. There seems to be a mistake in the Riemannian metric for the Poincaré ball.
10. In Equation (1), the formulation $\\mathbf{z}_i \\vee \\mathbf{z}_j$ is not defined. The dimension of $\mathbf{z}_i$ is not specified. The temperature-scaled softmax function is not defined.
11. No reference to CNN and transformers in line 482.
12. Notation:
    - the geodesic path \Gamma is sometimes bold and sometimes not
    - It should be  “Fréchet” mean instead of fréchet mean, and no reference is provided to it
    - The notation of the exponential map and the logarithmic map is not consistent - sometimes bold and sometimes not
13. The following ablation study is missing:
    - No LCA loss in the proposed model architecture
    - Same architecture using the hyperbolic LCA from Chami (2020) rather than the proposed one
    - Use the proposed LCA in Chami (2020)'s framework for hierarchical clustering
    - For HoroPCA++, isolate each change over HoroPCA to show the contribution
14. There are no visualization results of HoroPCA++.
15. Section 7 admits no large-scale testing, yet the paper claims scalability and “large datasets” as advantages.
16. There is no runtime or complexity analysis.
17. The appendix is hard to navigate (single section with A.1–A.7 spanning preliminaries, LCA, HoroPCA++, experiments, results, costs, encoders/decoders).
18. Is the work deep hyperbolic hierarchical clustering or deep hyperbolic clustering?
19. The paper assumes substantial prior knowledge and offers too little background and citation support. This makes the paper hard to follow.
20. No dataset references and statistics.
21. The provided code of the hyperbolic math library at https://anonymous.4open.science/r/hyperbolic-math-1002 is not found, and the use of code and models in https://anonymous.4open.science/r/hyperbolic-clustering-D73C is not straightforward
22. Use \cite{} inline for: line 31 (“Dasgupta’s cost by Dasgupta (2016)”), line 51 (“the seminal work of Chami et al. (2020)”), and similarly on lines 88, 99, 102, 176, 178, etc.
23. The reference or empirical support for: “implementing these operations is fraught with practical difficulties, including numerical instability … near-singular matrices during optimization” is missing
24. In line 93, n and d are not defined, and why $O(nd)$?
25. In line 222, manifold $\mathcal{M}$ is not defined and the formulation of the geodesic $\Gamma_{x,y}$ is also not defined
26. Line 292: Please justify the chosen pairwise similarity.
27. Eq. (4): how the hyperbolic LCA loss $\\mathcal{L}\_{LCA}$ enters $\\mathcal{L}\_{min}$ and $\\mathcal{L}\_{max}$? ​Please clarify how  $\\mathcal{L}\_{LCA}$ relates to $\\mathcal{L}\_{norm}$.
28. “Ours (Poincaré) + Spectral Clustering” and “Ours (Hyperboloid) + Spectral Clustering”: define the pipeline. Which space hosts the embeddings? Where is the affinity built?
29. Table 2–3 suggest the hyperboloid is better than the Poincaré in the proposed method. In such cases, why should one stick with the Poincaré model?
30. HoroPCA results in Table 1 are slightly lower than in its paper.
31. Current Fig. 3 shows a “moon-shape” that uses only a slice of the space. What causes this phenomenon?
32. Numerical instability was claimed in the hyperbolic dimensional reduction section, but there are no empirical results to show it.
33. The formulation of Dendrogram Purity and spectral NMI is missing - please give references and formulae. Also, please explain LR scheduler (Warmup-Cosine 20 epochs), activation (Swish), and curvature $c=0.1$. Why these choices?
34. HoroPCA++: the two adjustments are hard to follow (appendix A.3 and lines 329–344).
35. No sufficient background is provided for the deep clustering
36. On lines 428–429 (claims about “low” Dasgupta cost and “high” Dendrogram Purity): how low should a Dasgupta’s cost be considered to be as good, and how high should a Dendrogram Purity be considered good? There is no comparison to other methods.
37. The reported formats of Dendrogram Purity, NMI, and ARI are not consistent in the main paper and in the appendix.

**Questions:**

See Weaknesses above.

---

### Official Review · Reviewer_Znke · 2025-11-01

**Soundness:** 2
**Presentation:** 1
**Contribution:** 2
**Rating:** 2
**Confidence:** 5

**Summary:**

The paper introduces a deep hyperbolic hierarchical clustering framework to address limitations that have hindered hierarchical clustering in deep learning. Core contributions include: (1) a generalized, rectified definition of the hyperbolic lowest common ancestor applicable to both Poincaré Ball and Lorentz models under arbitrary curvature; (2) a scalable deep encoder that learns clusters in a very low-dimensional latent space relative to Euclidean baselines, improving efficiency on large datasets; and (3) HoroPCA++, a numerically stable hyperbolic dimensionality reduction method for lower-distortion visualization of learned hierarchies. The work targets geometric rigidity, scalability, and imprecise operations, positioning hyperbolic geometry as a natural fit for hierarchical structure.

**Strengths:**

The paper aims to study an important research topic of hierarchical clustering in hyperbolic space.

The proposed method demonstrate a superior performance in clustering tasks as shown in table 2.

**Weaknesses:**

Organization and presentation

Section 2 is overly verbose and provides limited coverage of recent literature.
Section 3 lists many concepts and formulas that are not used later in the paper; these can be omitted or moved to an appendix to improve focus and flow.
Figure 2 would be clearer if it showcases both the incorrect LCA point and the LCA defined by Definition 4.1 within the same figure for direct comparison.


Experimental evidence and claims

Table 1 shows no statistically significant improvement of HPCA++ over HPCA, calling into question the practical benefits of the proposed method.
I agree on the statement about the numerical stability issues with HPCA, the claim that HPCA++ resolves these issues lacks concrete evidence. A synthetic counterexample where HPCA fails but HPCA++ succeeds is needed to substantiate the claim.

Contribution clarity and novelty

The stated applicability to both the Poincaré ball and hyperboloid models appears straightforward and, on its own, is unlikely to constitute a substantive contribution.
The extension of HoroPCA to hyperbolic spaces with arbitrary curvature is straightforward; the paper does not convincingly articulate what is novel.
The purported numerical-stability advantage of HoroPCA++ over HoroPCA is not demonstrated through targeted examples or experiments.

Experimental setup and reporting

Baselines, datasets, and evaluation metrics in Section 6 are insufficiently specified and/or not properly cited, limiting the reproducibility and interpretability of the results.

**Questions:**

please refer to questions

---

### Note · Authors · 2025-12-03

**Comment:**

After careful consideration of the reviews, we have decided to withdraw our paper. However, as these reviews and our manuscript will remain part of the public record on OpenReview, we are providing this statement to clarify certain technical aspects of our work for future readers and to outline our roadmap for the next iteration of this research.
1. Clarifications on Technical Misconceptions We observed that some reviews contained fundamental misunderstandings regarding the mechanics of our proposed framework. We wish to correct the record on two specific points:

     * Hierarchy Generation: A reviewer stated that our method "does not produce a hierarchy." Our framework generates hierarchies by first encoding data into a hyperbolic latent space optimized via our continuous Dasgupta cost, followed by agglomerative clustering (e.g., Ward linkage) on the resulting hyperbolic embeddings. The resulting hierarchies are explicitly evaluated and reported in Table 3.
    * The "Wrong LCA" Issue: It was suggested that the incorrect formulation of the Hyperbolic Lowest Common Ancestor (LCA) in prior work is "not an issue." We respectfully disagree. Wang & Wang’s (2018) reformulation of Dasgupta’s cost – and consequently both Chami et al.’s (2020)’s and our LCA loss – is based on triplet samples in binary trees. In any binary tree, three nodes always induce a hierarchy i.e., one node must be an ancestor. Therefore, the LCA rectification is necessary to reflect the actual imposed hierarchy. While we create "leaf" embeddings, the training process establishes an internal hierarchy between latent representations that is explicitly constructed at inference time. Without our rectification, the loss function would fail to capture the true hierarchical relationships established during training, leading to suboptimal embedding quality. The correction applies in approximately 17% of all cases.
    * Generalization to Arbitrary Curvature: Some feedback suggested that extending our framework to arbitrary curvatures is "straight-forward" or trivial. We acknowledge that the theoretical concept may appear straightforward; however, deriving explicit closed-form solutions for geodesic orthogonal projections (Propositions 4.2 & 4.3) was a necessary step to establish a generalized framework that is numerically stable and practically efficient. Our ablation studies (Section 6.3) demonstrate that model performance is highly sensitive to the curvature parameter c, with no single value performing best across all metrics. Consequently, formally deriving these closed formulations for the LCA and HoroPCA++ is a strict prerequisite for optimizing this hyperparameter and enabling future work on learnable curvature.

2. Acknowledgment of Feedback and Future Directions We thank the reviewers for their constructive feedback, particularly regarding the evaluation landscape. We agree that comparing our method against broader, state-of-the-art self-supervised clustering baselines will strengthen the paper.
    * Modern Baselines: In the next version of this work, we will include comprehensive comparisons against modern deep clustering methods to better contextualize our performance beyond the standard deep clustering baselines (DEC, IDEC, DCN) used in this submission.
    * Large-Scale Hierarchical Datasets: We acknowledge the feedback regarding the need for testing on larger-scale data. While our current work validates the geometric foundations on standard benchmarks (MNIST, GTSRB), we recognize the importance of demonstrating scalability on datasets with rich, intrinsic natural hierarchies. Future evaluations will prioritize large-scale hierarchical benchmarks such as ImageNet and iNaturalist.
We thank the reviewers for their time and effort. We are confident that incorporating these changes will result in a stronger contribution to the field of hyperbolic representation learning.

**Withdrawal Confirmation:**

I have read and agree with the venue's withdrawal policy on behalf of myself and my co-authors.